# Wide Fontanels, Delayed Speech Development and Hoarse Voice as Useful Signs in the Diagnosis of KBG Syndrome: A Clinical Description of 23 Cases with Pathogenic Variants Involving the *ANKRD11* Gene or Submicroscopic Chromosomal Rearrangements of 16q24.3

**DOI:** 10.3390/genes12081257

**Published:** 2021-08-17

**Authors:** Anna Kutkowska-Kaźmierczak, Maria Boczar, Ewa Kalka, Jennifer Castañeda, Jakub Klapecki, Aleksandra Pietrzyk, Artur Barczyk, Olga Malinowska, Aleksandra Landowska, Tomasz Gambin, Katarzyna Kowalczyk, Barbara Wiśniowiecka-Kowalnik, Marta Smyk, Mateusz Dawidziuk, Katarzyna Niepokój, Magdalena Paczkowska, Paweł Szyld, Beata Lipska-Ziętkiewicz, Krzysztof Szczałuba, Ewa Kostyk, Agata Runge, Karolina Rutkowska, Rafał Płoski, Beata Nowakowska, Jerzy Bal, Ewa Obersztyn, Monika Gos

**Affiliations:** 1Department of Medical Genetics, Institute of the Mother and Child, Kasprzaka 17a, 01-211 Warsaw, Poland; jennifer.castaneda@imid.med.pl (J.C.); jakub.klapecki@imid.med.pl (J.K.); artur.barczyk@imid.med.pl (A.B.); olga.malinowska@imid.med.pl (O.M.); aleksandra.landowska@imid.med.pl (A.L.); tomasz.gambin@imid.med.pl (T.G.); katarzyna.kowalczyk@imid.med.pl (K.K.); barbara.wisniowiecka@imid.med.pl (B.W.-K.); marta.smyk@imid.med.pl (M.S.); mateusz.dawidziuk@imid.med.pl (M.D.); katarzyna.niepokoj@imid.med.pl (K.N.); magdalena.paczkowska@imid.med.pl (M.P.); pawel.szyld@imid.med.pl (P.S.); beata.nowakowska@imid.med.pl (B.N.); jerzy.bal@imid.med.pl (J.B.); ewa.obersztyn@imid.med.pl (E.O.); monika.gos@imid.med.pl (M.G.); 2Pediatric Surgery Clinic, Institute of the Mother and Child, Kasprzaka 17a, 01-211 Warsaw, Poland; maria.boczar@vp.pl; 3Unit of Anthropology, Institute of the Mother and Child, Kasprzaka 17a, 01-211 Warsaw, Poland; ewa.kalka@imid.med.pl; 4Department of Clinical Genetics and Pathomorphology, University of Zielona Góra, 65-417 Zielona Góra, Poland; aleksandra.pietrzyk@uz.zgora.pl; 5Rare Diseases Centre, Medical University of Gdańsk, 80-210 Gdańsk, Poland; b.lipska@gumed.edu.pl; 6Clinical Genetics Unit, Department of Biology and Medical Genetics, Faculty of Medicine, Medical University of Gdańsk, 80-210 Gdańsk, Poland; 7Department of Medical Genetics, Warsaw Medical University, 02-106 Warsaw, Poland; krzysztof.szczaluba@wum.edu.pl (K.S.); krutkowska@wum.edu.pl (K.R.); rploski@wp.pl (R.P.); 8Genetic Counselling Unit Kostyk&Kruczek, 31-436 Kraków, Poland; ewa@kostyk.pl; 9Genetic Department, Ludwik Rydygier Collegium Medicum in Bydgoszcz, Nicolaus Copernicus University of Toruń, 85-067 Bydgoszcz, Poland; agata.run@wp.pl; 10Genetic Counselling Clinic, Antoni Jurasz University Hospital, 05-094 Bydgoszcz, Poland

**Keywords:** KBG syndrome, *ANKRD11* gene, 16q24.3, dysmorphic syndrome, wide, delayed closing fontanels, hoarse voice, speech delay, psychomotor hyperactivity, short stature

## Abstract

KBG syndrome is a neurodevelopmental autosomal dominant disorder characterized by short stature, macrodontia, developmental delay, behavioral problems, speech delay and delayed closing of fontanels. Most patients with KBG syndrome are found to have a mutation in the *ANKRD11* gene or a chromosomal rearrangement involving this gene. We hereby present clinical evaluations of 23 patients aged 4 months to 26 years manifesting clinical features of KBG syndrome. Mutation analysis in the patients was performed using panel or exome sequencing and array CGH. Besides possessing dysmorphic features typical of the KBG syndrome, nearly all patients had psychomotor hyperactivity (86%), 81% had delayed speech, 61% had poor weight gain, 56% had delayed closure of fontanel and 56% had a hoarse voice. Macrodontia and a height range of −1 SDs to −2 SDs were noted in about half of the patients; only two patients presented with short stature below −3 SDs. The fact that wide, delayed closing fontanels were observed in more than half of our patients with KBG syndrome confirms the role of the *ANKRD11* gene in skull formation and suture fusion. This clinical feature could be key to the diagnosis of KBG syndrome, especially in young children. Hoarse voice is a previously undescribed phenotype of KBG syndrome and could further reinforce clinical diagnosis.

## 1. Introduction

KBG syndrome (MIM #148050) is a neurodevelopmental disorder characterized by short stature, macrodontia, developmental delay, behavioral problems such as psychomotor hyperactivity, velopharyngeal insufficiency causing feeding problems and speech delay, and skeletal anomalies presenting as delayed closing of fontanels. Currently, more than 160 patients have been described but only a few reports paid attention to less common features of KBG syndrome, with a prevalence of less than 30%, such as undescended testes, cardiac defects, ocular findings including high myopia, strabismus, cataracts, advanced puberty, skin and hair anomalies, juvenile idiopathic arthritis, oral frenula, tics, lipoma of corpus callosum, prominent and elongated coccyx [1,2,3,4,5,6]. Among the reported cases, wide anterior fontanel with delayed closure has been described only in 8 out of 44 examined patients (18.2%) [3,7]. Voice disturbances such as dysphonic voice were found in two patients described by Gnazzo et al. [6]. No patient with hoarse voice and low timbre has yet been described, although the term “dysphonic”, “hypernasal speech” and “speech disorders” can probably describe a similar voice change or abnormal phonation [3,4,6]. 

Most patients with KBG syndrome possess a pathogenic variant involving the *ANKRD11* (ankyrin repeat domain-containing protein 11) gene, which was reported as a causative gene in 2011 [8]. Chromosomal rearrangements encompassing the *ANKRD11* gene, mainly 16q24.3 deletion, cause a similar phenotype. In addition, clinical features typical of the KBG syndrome were also described in two siblings and their mildly affected mother, in whom an *ANKRD11* intragenic duplication was identified [7,9,10,11,12,13,14,15].

Herein, we describe a detailed clinical presentation of KBG syndrome in 23 patients: 13 with a pathogenic single nucleotide variant (SNV) in the *ANKRD11* gene and 10 with a 16q24 rearrangement encompassing the *ANKRD11* gene (nine patients with a microdeletion and one with a microduplication).

## 2. Materials and Methods

### 2.1. Patients and Clinical Investigations

All patients (23) were clinically examined and recruited by clinical geneticists. Most patients (17/23) were from a single genetic center—the Department of Medical Genetics in the Institute of Mother and Child in Warsaw—while 6 patients were from Genetic Departments in other major cities of Poland (Gdańsk, Kraków, Bydgoszcz). All but two patients—who were Ukrainian—were of Polish origin. 

All of the patients’ parents gave their informed consent for their children to be included in the study and for the anonymous publication of the patients’ photographs (with or without full face exposure), film and voice recordings in a professional medical journal. 

The group of patients in the study consisted of 23 patients: 13 patients with a pathogenic SNV of the *ANKRD11* gene and 10 patients with rearrangements of 16q24.3 involving the same gene—nine patients with a microdeletion and one with a microduplication.

#### Anthropometric Studies of Children with KBG Syndrome

Basic anthropometric examination was performed in the majority of patients. In 11 patients (3 girls and 8 boys) aged from 4 months old to 15 years old and diagnosed with KBG syndrome in the years 2016–2021, detailed anthropometric analysis was performed in the Anthropology Unit of the Institute of Mother and Child in Warsaw.

The children were measured using the previously described and approved technique of Martin and Saller (1957) and using standardized anthropometric equipment [16,17]. Four basic somatic measurements were collected from every evaluated child included in the study: height (length in the case of neonates and small babies), weight, head circumference and chest circumference. BMI score was also calculated.

Moreover, three basic head measurements were recorded in all children: length, width and circumference. An additional six cephalometric measurements were obtained from four boys. The definitions of all nine head measurements are as follows:
(1)Head circumference—measured with a tape around the maximum circumference of the head, just above the brow ridges at the front and the most protruding portion of the back of the head (occiput).(2)Head length (g—op)—from front to back—the distance from the brow ridge (glabella) to the most distant point on the back of the head on the occipital bone (opisthocranion) measured with a spreading caliper.(3)Head width (eu—eu)—the greatest distance between the most lateral points at the height of the parietal bones (euryon—euryon) measured with a spreading caliper.(4)Face breadth (zy—zy)—the greatest distance between the even points located most laterally on the zygomatic arches (zygion—zygion) measured with a spreading caliper.(5)Face height (n—gn)—the linear distance between the depression at the top of the nose (nasion) and the gnathion (on the lower edge of the mandible) measured with a sliding caliper.(6)Nose length (n—sb)—the distance from the depression at the top of the nose (nasion) to the subnasal point (at the base of the nose) measured with a sliding caliper.(7)Nose breadth (al—al)—the distance between the most lateral points of the wings of the nose (alare) measured with a sliding caliper.(8)Interocular breadth (en—en)—the distance between the inner corners of the eyes measured with a sliding caliper.(9)Biocular breadth (ex—ex)—the distance between the outer corners of the eyes measured with a sliding caliper.

The data analyzed in this paper were height, circumference and cephalometric features, measured with an accuracies of up to 0.001 m and weights of up to 0.1 kg.

The method of normative characteristics was used to analyze the differences in phenotypes. All analyzed data were standardized relative to means and standard deviations according to age and gender. The previously published values for the group of children from Warsaw were used as a reference [18]. Therefore, the standard deviations scores reflect the differences between measurements of somatic features in KBG patients when compared with healthy children. 

### 2.2. Cytogenetic and Molecular Investigations

In all patients with rearrangements of 16q24.3 encompassing *ANKRD11*, genomewide array analysis was performed using genomic array platform aCGH (Array Comparative Genomic Hybridization, array CGH).

An array CGH was performed using 60 K microarrays 8 × 60 K from Oxford Gene Technology (CytoSure ISCA, v3, Oxford, UK). The array used in this study contains 51,317-mer oligonucleotides probes covering the whole genome with an average spatial resolution of 60 Kb. DNA denaturation, labelling and hybridization were performed according to the manufacturer’s instruction. Genomic DNA was labelled for 2 h using the CytoSure Labelling Kit (Oxford Gene Technology, Oxford, UK), with no enzyme digestion. Hybridization was performed from 24 to 48 h in a rotator oven (Agilent) at 65 °C. Washing of arrays was performed using Agilent wash solutions 1 and 2. Arrays were scanned using an Agilent Technologies microarray scanner, followed by calculation of signal intensities using Feature Extraction software (Agilent Technologies). All scanned images were quantified using Agilent Feature Extraction software (V10.0, Agilent, Santa Clara, USA). All genomic coordinates are based on reference genome (NCBI37/hg19) and, for this article, converted to hg38. Data analysis was performed using the CytoSure Interpret Software (Oxford Gene Technology) and the circular binary segmentation algorithm. The calling thresholds were deviations of a circular binary segmentation (CBS) segment from zero log ratio of +0.30 for duplications and −0.5 for deletions. The results were then classified with CytoSure Interpret Software (V4.11.40, Oxford Gene Technology, Oxford, UK). The quality control metrics were monitored with CytoSure Interpret software (V4.11.40, Oxford Gene Technology. Oxford, UK). 

The microarray used in this analysis did not contain SNP probes, and it did not detect polyploidy, inversion, balanced translocation and regions of absence of heterozygosity. 

In 11 patients, a targeted sequencing analysis was performed. The tested genes included those related to dysmorphic syndromes and craniosynostosis (detailed list available upon request). The experiments were performed using SeqCap EZ HyperCap Workflow (version 2.3, Roche, Madison, WI, USA). The genomic DNA (100 ng) was extracted from fresh blood samples and fragmented using a KAPA Hyper Plus enzymatic fragmentation protocol. The library was enriched using custom SeqCap EZ Choice Enrichment Panel—the procedure was performed according to the manufacturer instructions. Libraries were sequenced on MiSeq or NextSeq550 sequencers (2 × 75 cycles, Illumina, San Diego, CA, USA). For two patients, exome sequencing was performed—one was sequenced in CeGaT GmbH (Tubingen, Germany; library: Twist Human Core Exome Plus; sequencing: NovaSeq 6000) and the other one was sequenced in the Department of Genetics of Warsaw Medical University (library: Illumina TruSeq Exome; sequencing: HiSeq1500). Data analyses for both panel and exome sequencing was performed using data analysis pipeline developed in the Institute of Mother and Child in Warsaw. The reads were aligned to the GRCh38 reference genome using BWA MEM. The Genome Analysis Toolkit 4 was used to identify single nucleotide variants and small insertion/deletions. Variant calling was performed with the Haplotype-Caller. Variant annotation was carried out with Ensembl Variant Effect Predictor (version 101, EMBL-EBI, Hinxton, UK) using resources from ClinVar clinical database, gnomAD v3 (Broad Institute, Cambridge, MA, USA), and an in-house population database and the dbNSFP v4.1 (University of South Florida, Tampa, FL, USA) functional predictions database [19,20,21,22]. The quality data are presented in Appendix A.

Sanger sequencing was used to confirm the presence of the variant and to perform segregation studies in family members.

## 3. Results

### 3.1. Clinical Results

The clinical characteristics of our group of patients are listed in Table 1.

All patients with micro-rearrangements of 16q24.3 were diagnosed accidentally without prior suspicion of KBG syndrome. An aCGH analysis was performed in these patients due to other indications such as developmental delay or intellectual disability with congenital defects and/or short stature and dysmorphic features. Moreover, all but one patients (Patient 13) (12 out of 13) with a pathogenic variant in the *ANKRD11* gene were tested earlier with aCGH, revealing no genomic imbalance. Nine of 13 (69%) patients tested with next-generation sequencing technology were previously suspected to have KBG syndrome; thus, the *ANKRD11* gene was analyzed by the first choice among the genes available for testing in the gene panel. The majority of identified variants were novel and were loss of function mutations. In one patient, the identified missense variant c.7607G>A (p.Arg2538Gln) was not found in population databases (gnomAD v3, in-house database with >1000 exomes) but was predicted to be pathogenic by predictive algorithms and found to occur de novo after family testing.

All patients demonstrated dysmorphic facial features, some of which are shown in the photos of a subset of the patients in Figure 1.

### 3.2. Results of Anthropometric Studies of Children with KBG Syndrome

The measurements of the basic somatic features of all of the children included in the study, their standardized and mean values are featured in Table 2.

The average age of patients harboring a mutation in the *ANKRD 11* gene was 8.4 years (range 0.3–26 years). Similarly, the average age of patients with micro-rearrangements of 16q24.3 was 8.3 years (range 3.8–20 years). In contrast, the average age of patients with wide fontanel and mutation of the *ANKRD11* gene was lower: 6.8 years. 

Compared with the reference group, the children diagnosed with KBG syndrome in this study had negative mean deviations for height, weight, BMI score, and head and chest circumference (Table 2). A clinically consistent feature was thus that the patients are shorter, have lower body weight and have decreased head and chest circumferences compared with unaffected children. The greatest differences were identified in the height and head circumferences, with mean standard deviation of −1.14 and −1.20, respectively. All mean deviations were within the normal range (+/−2.0 SDs).

However, in some of the affected children, deviations in the height and head circumferences were below the lowest reference range (−2.0 SDs). These decreased values were identified more often in younger patients below the age of 7, while all measurements were within the normal range (+/−2.0 SDs) in children above 7 years of age. 

Cephalometric characteristics underwent further analysis. Figure 2 shows the standard deviations of the obtained basic head measurements: head circumference, length (g—op) and width (eu—eu).

The head measurements of the patients were narrower (−0.87 SDs), longer (0.10 SDs) and smaller in circumference (−1.20 SDs) in comparison with the reference group. The standard deviations of the assessed data were within the normal range.

Figure 3 displays the range of detailed features of the heads of male patients affected by KBG syndrome. When compared with the reference group, children with KBG syndrome were characterized by the following:
−Increased head length (g—op) and facial morphologic length (n—gn);−Wider face (zy—zy);−Increased interocular distance—inner canthal (en—en) and outer canthal (ex—ex);−Decreased head circumference and width (eu—eu); and−Shorter (n—sn) and narrower (al—al) nose.

Increased standard deviation exceeding the upper limits of the normal narrow range (+1.0 SDs) was noted only in two measured variables: the greatest face width (zy—zy) and the inner canthal distance (en—en), with standard deviations of 1.29 SDs and 1.05 SDs, respectively. All other head measurements were within the normal narrow range (+/−1.0 SDs).

### 3.3. Unexpected Results

Patients with unexpected findings, with a negative *ANKRD11* analysis and without deletion/duplication of the 16q24.3 region as well as patients with interesting findings in aCGH are presented in Table 3. 

Three patients without *ANKRD11* mutation, suspected earlier to have KBG syndrome, were diagnosed with different rare genetic syndromes (Figure 4). Each of these patients had some features overlapping with KBG syndrome. In the case of the patient diagnosed with Weiss–Kruszka syndrome, dysphonic voice, intellectual disability and psychomotor hyperactivity were observed. Short stature, feeding problems and dysmorphic features were found in the patient with Keipert syndrome, while hoarse voice, psychomotor hyperactivity, intellectual disability and hirsutism were present in the female with Pierpont syndrome. None of these patients had a history of wide anterior fontanel in early childhood. 

Two patients (Patient 20 and Patient 21) with microdeletions of 16p24.3 localized in the first intron of the *ANKRD11* gene had similar phenotypes: short stature below −3.3 SDs, dysmorphic features with long nose and elongated face, learning problems and psychomotor hyperactivity. One of these patients had macrodontia. The other one had severe brachydactyly, and additional analyses revealed the presence of a variant of unknown significance NM_014112.4:c.2834C>T (p.(Pro945Leu) in the *TRPS1* gene (Figure 5). In both cases, an analysis of the parent DNA was not possible; therefore, we could not determine the pathogenicity of the microdeletion and the SNV in the *TRPS1* gene.

Patient 17 with a duplication of 16q24.3 that occurred de novo presented with behavioral disorders of the autistic spectrum and speech delay.

In one patient (Patient 12), apart from a mutation in the *ANKRD11* gene, a deletion of 1q21.1 inherited from the unaffected mother was identified. 

## 4. Discussion

It has been demonstrated previously that KBG syndrome is characterized by short stature and distinctive facial features [3,23]. However, these characteristics are often mild and nonspecific, and none of them are a prerequisite for the KBG syndrome diagnosis [8]. 

The results of the anthropometric measurements demonstrated that children with KBG syndrome had smaller values of investigated body dimensions. They were characterized by short height and low weight as well as by decreased head width (eu—eu) and circumference. The faces of affected children were wider (zy—zy), and increased inner and outer canthal distances were also observed. An evaluation of the unusually shaped head and face with characteristic though not pathognomonic facial features may thus be used to raise suspicion of this disorder and to consider further genetic testing to confirm diagnosis.

In the first years of life, the clinical diagnosis of KBG syndrome is particularly difficult because of the usually mild facial phenotype and the lack of cardinal clinical signs such as macrodontia of upper central incisors, which is usually noted in permanent teeth. Sixty-nine percent of our patients under 7 years of age with mutations of the *ANKRD11* gene presented with delayed closure of the wide anterior fontanel. In two patients diagnosed at the age of 4 months, apart from mild dysmorphic features, wide anterior fontanels were the only pathologic feature found (Figure 6).

In the group of nine patients with wide and delayed closure of anterior fontanels and mutations in the *ANKRD11* gene, 7/9 patients (78%) had a suspicion of KBG syndrome.

As noted by Low et al., delayed closure of the anterior fontanel is one of the most distinctive clinical features of *ANKRD11* mutation patients according to the Decipher gene browser; this may probably contribute to the characteristic face shape in KBG syndrome [3]. There are but a few disorders associated with a wide fontanel of delayed closure or with a combination of a wide fontanel and a broad forehead—examples are cleidocranial dysplasia, Robinow syndrome and achondroplasia (Table 4). However, aside from a broad forehead, no other facial features of patients with these syndromes are similar to those of KBG syndrome patients. There are probably other facial bone and suture defects that contribute to the triangular face shape in KBG syndrome apart from a wide fontanel.

Many genetic syndromes are associated with voice disturbances, such as Cat-cry syndrome, Di George syndrome, Williams syndrome, Smith–Magenis syndrome, some metabolic disorders and congenital hypothyroidism. Voice hoarseness appears to be the most frequent voice pathology in genetic syndromes. We suggest that KBG syndrome should be added to the group of well-known syndromes associated with voice disturbances (Table 5). To our knowledge, hoarse or dysphonic voice has not been previously described in Weiss–Kruszka or Pierpont syndrome. To ascertain if this can be one of the diagnostic criteria in these syndromes, a larger set of patients should be assessed for voice disturbances. 

A hoarse voice can be described as a voice that sounds breathy, raspy or strained and could be softer in volume or lower in pitch. Hoarseness is often a symptom of pathology in the vocal folds of the larynx. The capacity to observe voice disturbances in patients requires the clinical geneticist to have some sensibility to sound (fortunately, not necessarily perfect pitch) and the ability to focus attention on the patient’s voice.

Hoarseness of voice was observed in more than half of our cohort of patients and was especially astonishing in children in their first years of age (Audio Recording 1). It was heard mildly in two of our patients almost from their first sounds, before their first year of age (Audio Recording 2, DVD 1). We emphasize that the presence of this feature in KBG syndrome patients is specific, and it seems that hoarse voice can be useful in making an initial diagnosis of this disease.

We suggest that both a wide anterior fontanel resulting from delayed suture closure as well as hoarse voice should be included among the major clinical criteria of the diagnostic aid for KBG syndrome presented by Low et al. (Table 6), together with psychomotor hyperactivity and problems in gaining weight despite good appetite. According to Low et al., one of the major criteria in diagnosing KBG syndrome is recurrent otitis media and/or hearing loss. However, we did not observe this feature in our cohort of patients. This could perhaps be explained by the younger age of our patients—(4 months–26 years) compared with the patients presented by Low et al. (2 year–47 years). Regarding minor criteria proposed by the same author, the majority of our patients had brachydactyly, but autism was diagnosed only in two patients, seizures were diagnosed in one patient, and no palate anomalies were observed in any patient. 

The most frequent clinical features in our cohort of KBG syndrome patients were psychomotor hyperactivity and speech delay, which were present together with dysmorphic features typical for this disease. However, inquiring on the period of anterior fontanel closure and paying attention to voice timbre could be the clue to the right diagnosis. 

These criteria seem to be especially useful in children below 7 years of age when macrodontia was not yet apparent. Furthermore, in children below 2 years of age, a wide anterior fontanel and a hoarse voice could be the only signs of KBG syndrome as dysmorphic features at this age are usually mild. Finally, KBG syndrome should be included in the differential diagnosis of children with wide anterior fontanel of delayed closure even in the absence of other symptoms, especially in the first year of life.

The fact that none of the three patients previously suspected of KBG syndrome but did not present *ANKRD11* mutation had a history of wide anterior fontanel in early childhood and that two of them had a hoarse or dysphonic voice suggests that the feature of a wide fontanel is more specific in diagnosing KBG syndrome than voice disturbances. However, a hoarse voice could be an additional useful sign in diagnosing KBG syndrome when it coexists with a wide anterior fontanel or other major signs of the syndrome.

In Patient 22 with partial sagittal craniosynostosis confirmed by CT, partial deletion of exon 3 of the *ANKRD11* gene was detected by aCGH. To our knowledge, craniosynostosis has not been described in 16q24.3 rearrangements encompassing the *ANKRD11* gene thus far. The patient needs further diagnosis, but it is known that wide sutures and fontanels with delayed closure are rare clinical expressions of disturbances in cranial sutures development in craniosynostosis syndromes, such as in Saethre–Chotzen syndrome associated with *TWIST1* gene mutations or Apert syndrome associated with *FGFR2* gene mutations [24,25]. It is possible that, in KBG syndrome, in which delayed closure of a wide fontanel is a common feature, craniosynostosis could result from disturbances of cranial suture development associated with the *ANKRD11* gene. The fact that wide, delayed closing fontanels were observed in more than a half of our patients with KBG syndrome confirms the role of the product of *ANKRD11* gene in skull formation and suture fusion.

Duplications of 16q24.3 encompassing the *ANKRD11* gene have been rarely reported in the medical literature [15]. Crippa et al. described a family with intragenic duplication and mild KBG phenotype with short stature, dysmorphic features, macrodontia, nasal voice, moderate intellectual disability as well as urinary system defects. Our patient (Patient 17) has a de novo duplication of 16q24.3 that encompassed three genes: *SLC22A31, ZNF778* and *ANKRD11.* This duplication was interpreted as a variant of unknown pathogenicity. The clinical features of this patient included psychomotor hyperactivity, autism spectrum disorders, speech delay, scoliosis and mild dysmorphic features.

The clinical phenotype associated with a mutation, deletion, partial deletion or duplication of the *ANKRD11* gene and other clues, e.g., gene constraint metrics in gnomAD, suggests that it is a dosage-sensitive gene critical for normal development affecting especially the central nervous system and the skeletal system, including skull development. Additional studies are required to fully understand the role of the *ANKRD11* gene in human development.

## Figures and Tables

**Figure 1 genes-12-01257-f001:**
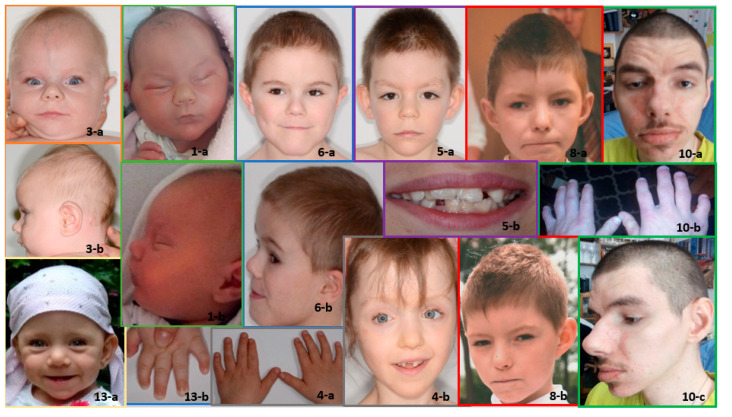
Dysmorphic features in a subset of patients with KBG syndrome. Note the wide and triangular face shape (in all patients) and pear-shaped skull (Patient 1-a and 8-a), narrow (Patient 6-a and 8-a) and elongated palpebral fissures (Patient 1-a, 4-b, 5-a and 8-a ), short nose with broad tip (Patient 1-a, 3-a, 5-a and 6-a), narrow upper lip (Patient 4-b, 5-a, 6-a, 8-a, and 13-a), brachydactyly (Patient 10-b and 13-b) and clinodactyly of the fifth digit (Patient 4-a, 10-b and 13-b). The phenotype changes with age—the cases presented here are from ages 4 months (Patient 1 and 3) to 26 years (Patient 10). The oldest patient has small palpebral fissures, unilateral ptosis, a big nose, a thick lower lip and marked interphalangeal joints (Patient 10-a-c).

**Figure 2 genes-12-01257-f002:**
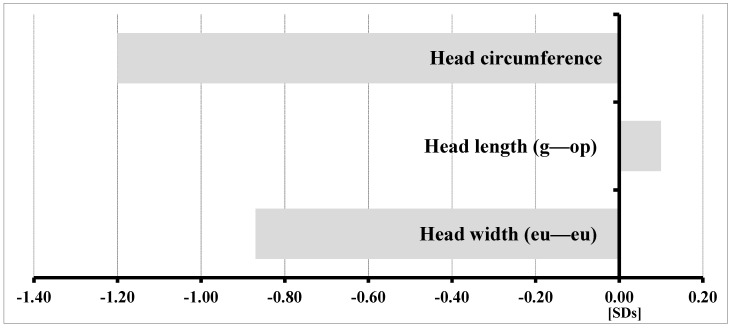
Profile of the mean normalized head characteristics of all examined children with KBG syndrome.

**Figure 3 genes-12-01257-f003:**
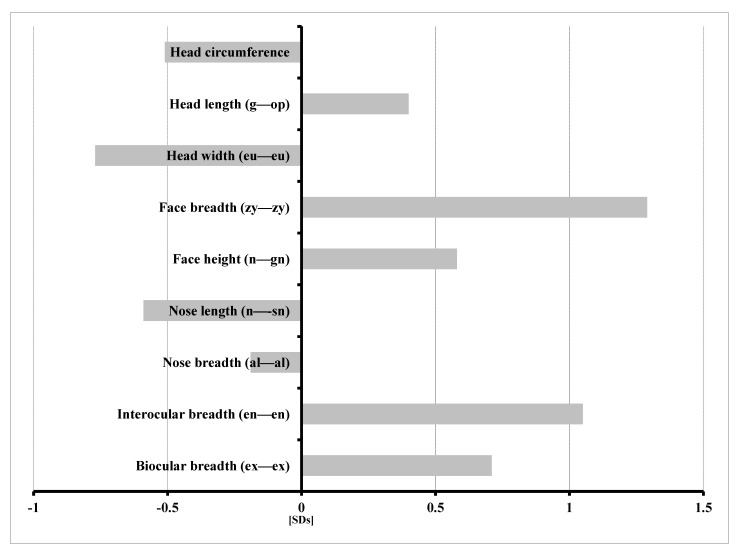
Profile of mean normalized cephalometric features of four boys with KBG syndrome.

**Figure 4 genes-12-01257-f004:**
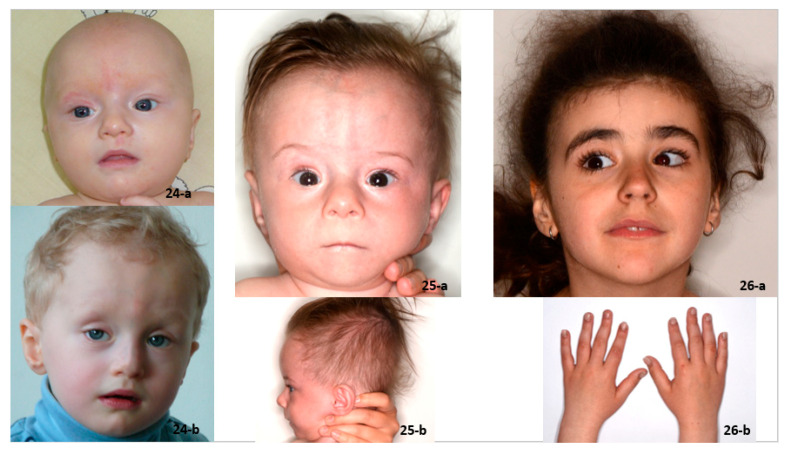
Facial features of patients suspected to have KBG syndrome without *ANKRD11* mutations: Patient 24-a,b—Weiss-Kruszka syndrome; Patient 25-a,b—Keipert syndrome; and Patient 26-a,b—Pierpont syndrome. In all patients, note the elongated palpebral fissures and short nose, which were the bases of KBG syndrome suspicion, apart from other clinical signs.

**Figure 5 genes-12-01257-f005:**
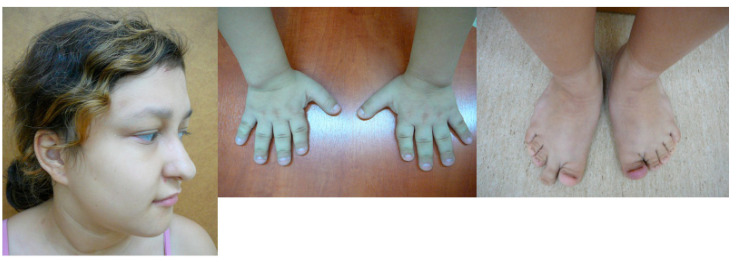
Patient 20 with short stature below −3.3 SDs, with microdeletion in the *ANKRD11* gene localized in intron1 and with a variant in the *TRPS1* gene of unknown pathogenicity.

**Figure 6 genes-12-01257-f006:**
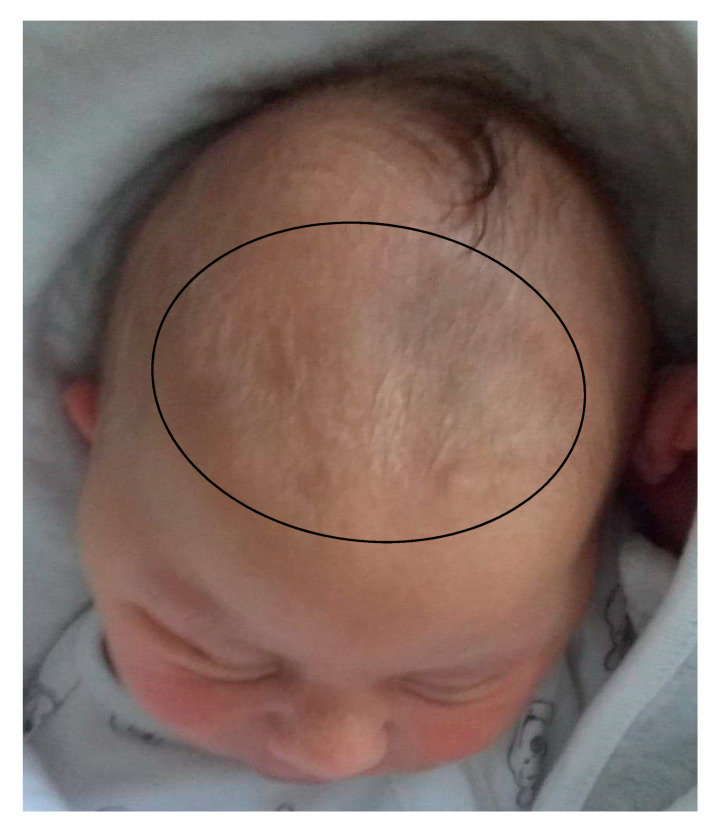
Neonate (Patient 1) with KBG syndrome and a very wide anterior fontanel.

**Table 1 genes-12-01257-t001:** Clinical characteristics of KBG patients.

Patient	KBG Gene Mutation/16q24 Rearrangement	Pathogenicity	Age of Diagnosis (Years)	Poor Weight Gain/Feeding Problems	Short Stature	Macrodontia	Hoarse Voice	Behavioral Problems	DD/ID	Delayed Speech	Wide Fontanel, Delayed Closure
**1**	c.1903_1907del (p.Lys635Glnfs*26)	Pathogenic (HGMD: CD150411)	0.3	−	+ (−2.14 SDs)	NA	+	NA	−	NA	+
**2**	c.1903_1907del (p.Lys635Glnfs*26)	Pathogenic (HGMD: CD150411)	2.6	−	− (−0.52 SDs)	+	+	−	−	+	+
**3**	c.7607G>A (p.Arg2538Gln)	Pathogenic (de novo)	0.3	+	− (−1.66 SDs)	NA	−	NA	−	NA	+
**4**	c.4558del (p.Asp1520Thrfs*11)	Pathogenic (de novo)	7	+	+ (−3.61 SDs)	+	+	+	−	+ 3 words	+
**5**	c.2395A>T (p.Lys799Ter)	Pathogenic (de novo)	4.4	+	− (−0.48 SDs)	−	+	+	Delayed motor development	+	+
**6**	c.1389dup (p.Gly464Argfs*29)	Pathogenic (de novo)	4	+	+	−	+	+	+	+	+
**7**	c. 7552C>T (p.Gln2518Ter)	Pathogenic (ClinVar: 280739)	10	−	+ (−2.5 SDs)	+	+	+	Learning problems	+	+
**8**	c.2828_2829del (p.Glu943Valfs*74)	Pathogenic	16	−	− (−1.59 SDs)	+	+	+	+	+	+
**9**	c.6340C>T (p.Gln2114Ter)	Pathogenic (de novo)	8	+	−	+	−	+	+	+	−
**10**	c.3295_3296del (p.Phe1099Leufs*2)	Pathogenic (de novo)	26	+	−	+	+	−	+	+	-
**11**	c.3771dup (p.Glu1258Argfs*25)	Pathogenic	8	+	+	ND	−	+	+ mild	+	−
**12**	c.1385_1388del (p.Thr462Lysfs*47)	Pathogenic (HGMD: CD1412989)	22	+	−	+	+	+	+ moderate ID	+	+
**13**	c.6053_6057del (p.Pro2018Argfs*12)	Pathogenic	3	+	polydactyly unilateral	+	−	+	DD epilepsy	+	+
**14**	arr[hg38]13q21.31(61968361-63533241)x1; 16q24.2q24.3(87921245-89417758)x1	Pathogenic	8	+	− (−1.02 SDs)	+	−	+	Learning problems	+	−
**15**	arr[hg38] 16q24.3(89171712-89274753)x1	Pathogenic (de novo)	5.5	−	− (−1.71 SDs)	−	+	−	IQ-nl	−	+
**16**	arr[hg38]16q24.3(89277485-89517986)x1	Unknown pathogenicity	3.5	−	−	−	−	+	DD	+	+
**17**	arr[hg38] 16q24.3(89195406-89489612)x3	Pathogenic	6	−	−	−	−	+ ASD	−	+	−
**18**	arr[hg38]16q24.3(89266045-89305443)x1	mosaic	5	+	− (−1.08 SDs)	−	+	+	−	−	−
**19**	16q24.3(89277485-89431539)x1	Pathogenic(de novo)	6	+	−	−	−	+	Mild ID	+	+
**20**	arr[hg38] 16q24.3(89458995-89487166)x1	Unknown pathogenicity	22	−	+ (−3.3 SDs)	−	−	+	Learning problems	−	ND
**21**	arr[hg38] 16q24.3(89481147-89489612)x1	Unknown pathogenicity	15	+	+ (−3.37 SDs)	+	−	+	Learning problems	+	−
**22**	arr[hg38] 16q24.3(89409759-89418313)x1	Likely pathogenic/Pathogenic	5	−	−	−	−	ASD	DD	+	Partial sagittal craniosynostosis
**23**	arr[hg38] 16q24.3(89277485-89489140))x1	Pathogenic	13	+	+	+	+	ADHD	Mild ID	_+_	−

Sequence variants were described according to NM_013275.6 and NP_037407.4 reference sequences. Abbreviations: DD, developmental delay; ID, intellectual disability, ASD, autism spectrum disorder; ADHD, attention deficit hyperactivity disorder, IQ, intelligence quotient; nl, normal; SDs, standard deviations; NA, not applicable; ND, no data; “+”, present; “−”, absent.

**Table 2 genes-12-01257-t002:** Anthropometric characteristics of children with KBG syndrome—measurement results and normalized values.

Patient	Age	Weight		Height (Length)		BMI		Head Circumference		Chest Circumference	
	Years	kg	Z-Score	cm	Z-Score	kg/m^2^	Z-Score	cm	Z-Score	cm	Z-Score
1	0.3	5.10	−1.75	58.6	−2.14	14.85	−1.04	38.9	−1.82		
3	1.4	11.3	−0.09	79.4	−0.77	17.80	0.58	45.7	−2.04	49.8	0.72
2	2.6	10.4	−2.50	91.0	−0.52	12.56	−3.13	47.1	−2.36	47.0	−1.76
5	4.4	16.4	−0.88	106.0	−0.48	14.60	−0.79	50.0	−1.27	51.6	−0.98
9	5.4	17.1	−0.93	111.0	−0.48	14.12	−0.89	48.6	−1.73	50.5	−1.23
6	5.9	18.5	−0.93	110.0	−1.48	15.29	−0.31	50.0	−1.58	55.0	−0.48
11	6.8	17.7	−1.89	112.5	−2.19	13.99	−1.15	48.4	−3.05	54.7	−1.30
15	7.2	21.9	−0.59	114.5	−1.51	16.73	0.24	50.5	−0.73	56.2	−0.37
17	7.3	26.8	0.16	125.1	−0.28	17.12	0.46	53.6	0.57	59.5	−0.05
7	11.3	40.8	−0.12	141.8	−1.13	20.29	0.59	56.7	1.48	73.2	0.50
8	14.8	53.4	−0.73	160.5	−1.59	20.34	0.00	55.0	−0.67	79.6	−0.08
	Mean z-score		−0.93		−1.14		−0.49		−1.20		−0.50

**Table 3 genes-12-01257-t003:** Unexpected results of genetic analyses performed in patients clinically suspected of the KBG syndrome.

Clinical Signs	Syndrome/Gene
Patient 24: Intellectual disability, dysphonic voice, dysmorphic features	Weiss–Kruszka syndrome *ZNF462*: NM_021224.6:c.4784_4785del; p.(His1595Leufs*9)
Patient 25: IUGR, short stature (−3.83 SDs), developmental and speech delay, dysmorphic features	Keipert syndrome *GPC4*: NM_001448.3:c.881C>T; p.(Ala294Val)
Patient 26: short normal stature (−1.83 SDs), hoarse voice, hirsutism, psychomotor hyperactivity.	Pierpont syndrome *TBL1XR1*: NM_024665.7: c.513A>C; p.(Glu171Asp)
**Unexpected results of aCGH:**	
**2 patients with 16q24.3 deletion (Patient 20 and Patient 21)**	
Short stature below −3.3 SDs Similar dysmorphic features (Figure 5): long nose, long face Learning problems, psychomotor hyperactivity Macrodontia in one patient Severe brachydactyly in one patient with a variant of unknown significance in the *TRPS1* gene: NM_014112.4:c.2834C>T (p.(Pro945Leu)	Both patients with microdeletions of unknown pathogenicity localized in intron 1 of the *ANKRD11* gene Analyses of parents’ DNA not performed
**Patient 17 with de novo 16q24.3 duplication**	
Autism spectrum disorder, stereotypic movements, speech delay	Duplication of: *SLC22A31, ZNF778* and *ANKRD11* of unknown pathogenicity

**Table 4 genes-12-01257-t004:** Disorders with wide, delayed closing anterior fontanel.

**Most Common**
Achondroplasia
Congenital hypothyroidism
Down Syndrome
**Less Common**
Skeletal Disorders
Cleidoclanial dysplasia
Acrocallosal syndrome
Campomelic dysplasia
Hypophosphatasia
Kenny-Caffey syndrome
Osteogenesis imperfecta
Dysostosis Stanescu type
**Dysmorphogenetic Syndromes**
KBG syndrome
Robinow syndrome
Beckwith-Wiedemann syndrome
Zellweger syndrome
Cutis laxa
VATER association
Otopalatodigital syndrome
Occipital horn syndrome
Autosomal recessive cutis laxa type 2A
Partial trisomy of the short arm of chromosome 9

**Table 5 genes-12-01257-t005:** Differential diagnosis of voice disturbances in genetic syndromes.

KBG syndrome—low pitched hoarse voice
Williams Syndrome—low pitched hoarse voice
Smith–Magenis syndrome—low pitched hoarse deep voice
Congenital Hypothyroidism—low pitched hoarse deep voice
Genetic metabolic diseases—mucopolysaccharidosis, Farber diseases, disseminated lipogranulomatosis, lipoid protenois, Morquio A
Cutis laxa (pendulous skin and hoarse cry)—vocal folds thickening
Dubowitz syndrome—high pitched hoarse voice
Idiopathic familial voice disorder—vocal fold paralysis
Ehlers–Danlos syndrome type VIII—hoarse voice
Costello syndrome—hoarse voice
Aicardi–Goutieres syndrome—low pitched hoarse voice
Werner syndrome—hoarse voice
X-linked hypohidrotic ectodermal dysplasia (XLHED)—hoarse raspy voice

**Table 6 genes-12-01257-t006:** Proposal of diagnostic aid in KBG syndrome (based on Low et al.).

Major Criteria	Proportion of Patients in this Cohort with Feature	Children below 7 Years of Age	Children below 2 Year of Age
Psychomotor hyperactivity/ADHD	85.7%	90%	-
Speech delay	80.9%	82%	-
Gain weight problems with good appetite	56%	80%	-
Height below the 10th centile	56%	40%	25%
Wide, delayed closing anterior fontanel	56%	69%	100%
Hoarse voice	56%	69%	75%
Macrodontia	52%	-	-
1st degree relative with KBG Syndrome	0.0%	no relatives available	

## Data Availability

Submission of the sequencing data to databases may be problematic for ethical reasons, but the data are available upon request to the corresponding author in compliance with EU GDPR.

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
