# Peer review of "Wide Fontanels, Delayed Speech Development and Hoarse Voice as Useful Signs in the Diagnosis of KBG Syndrome: A Clinical Description of 23 Cases with Pathogenic Variants Involving the ANKRD11 Gene or Submicroscopic Chromosomal Rearrangements of 16q24.3"

_genes, 2021, doi:10.3390/genes12081257_

Round 1
Reviewer 1 Report
Minor suggestions:
* p1 line 41 - no need for "next generation" description of sequencing, will help keep word count down for abstract.
* p1 abstract - please mention that KBG is an autosomal dominant disorder when caused by ANKDR11 mutations/microdeletions
* p1 line 43 - 56% had both delayed closure of fontanel and hoarse voice? Implies the same patients - even if numbers are the same, if it's not exactly the same set of patients, best to specify 56% for each characteristic separately.
* p2 line 48/49 - suggest rewording as "Hoarse voice is a previously undescribed phenotype of KBG syndrome and could reinforce clinical diagnostics."
* p2 line 74 - no need to identify 17 from 1 centre here, this info is more fully given immediately in the next paragraph of Materials & Methods
* p2 line 90 - Delete line "Clinical data including photographs, anthropometry were analyzed by authors". This is fully described in the next section.
* p2 line 95 - indicates 10 patients had microrearrangements of 16q24.3, but these are described more generally as rearrangements in the previous sentence. Please define the type of rearrangements more specifically.
* p2 line 94 - p3 line 96 - this information is repeated in the results section, can be deleted in methods
* p3 section 2.1 - I'm not familiar with the term somatotypes, is this equivalent to phenotypes?
* p4 line 169 - "according"
* p4 line 166-176 - Please provide all software versions and parameters, can be a short supplementary methods if space is limited.
* dbNSFP is mentioned on p6 line 192, this should be included in methods
* p 4/5/6 - Table 1 clinical characteristics. Please significantly expand the caption, e.g. explain "dn" (presumably de novo), what database are the mutation accessions referring to, what database is the likely pathogenic/pathogenic classification coming from. Formatting is v difficult to read. Please make it clear if 'likely pathogenic' is coming from in silico predictions such as for patient BB as described below.
* Table 1 patient NJ - "brak nazwy białkowej" = "no protein name"?
* p6 line 184 - how were they diagnosed "accidentally"? were they having the aCGH panel done for other reasons?
* p6 line 190 - which patient? give the 2 character id from Table 1.
* p6 Figure 1. I can't figure out which photos relate to which ids in parts of the figure - please separate these out into a/b/c/etc sub-figures so that all photos are grouped by individual.
* p6 line 198. Figure 1 only shows 6(?) patients, please say "a subsetof which are shown in Figure 1".
* p6 line 194-196 - paragraph on age should move into the next section after Table 2 which includes the raw data
* p7 Table 2 - add the patient ids as a column so readers can map the individuals to Table 1
* p7 line 211 - please change "healthy" to "unaffected".
* p8 Figure 2 - Either delete this figure or change to box & whiskers plot plus individual data points for each of the relevant metrics. As it is, it only repeats the summary line of Table 2.
* p8 Figure 3 and p9 Figure 4 - The data in both of these figures would be better presented as tables as in Table 2. If figures are still thought to be also needed, use a box and whiskers plot with the individual data points added (see https://www.r-graph-gallery.com/89-box-and-scatter-plot-with-ggplot2.html for examples).
* p13 Table 4. KGB?
Major suggestions:
* It's not something that can be changed now, but why is this all on hg19/GRCh37? hg38/GRCh38 was first released in 2013.
* p 10 Table 3. This should all be written up as paragraphs, a table isn't the best way to present these mini-case studies. Refer to each individual throughout by the ids from Table 1.
* All of the added in material from the journal appears to not have been completed by the authors.
* Sequencing data should be deposited at the European Genome/Phenome Archive if possible and an accession noted in the paper. Editor: note that national legal systems do prevent this in some cases, but the expectation is that authors make data as available as possible.
Reviewer 2 Report
This review describes the manuscript titled ‘Wide fontanels, delayed speech development and a hoarse voice as useful signs in the diagnosis of KBG syndrome. Clinical description of 23 cases with pathogenic variant involving the ANKRD11 gene or submicroscopic chromosomal rearrangements of 16q24.3.’
The authors present the clinical presentation of 23 patients with KBG syndrome, which is a rare neurodevelopmental disorder (160 patients previously described) linked to mutation of the ANKRD11 gene. They include genetic mutation analysis using exome sequencing and array CGH. They dscribe new and previously under-reported phenotypic features of the disease including delayed speech, poor weight gain, delayed closure of fontanel and hoarse voice.
The information in this manuscript is novel and usefull to the medical and genetic disease community – with 23 new cases described adding a substantial proportion to the only 160 previously described. However I don’t feel the manuscript presents the results in a clear and concise manner, there are multiple spelling and grammatical errors, as well as poor layout in the tables and figures. Furthermore the results for antropometric characterisitcs are described as mean z-scores across each category, using a reference population of healthy controls as a comparison. For all except two physical characteristics (the greatest face width (zy-zy) and the inner canthal distance (en-en) in males) the deviation in z-score were within the normal range of standard deviation. Therefore there is not sufficient data to demonstrate a significant deviation from normal range of phenotypic characteristics.
I have made a few examples of spelling/formatting errors that could be rectified, but this is not exhaustive and I advise the authors to review the manuscript thoroughly for such errors.
- Numerous spelling an grammatical errors throughout (e.g. Table 1: pacjent -> patient; wieght -> weight; table 2 ( Lenght -> Length)
- Table 1 is hard to read – change layout so headings are easier to read, or shorter, formatting of the whole table should be addressed – I am not sure if this is an error in translation when saved for peer review. There is also some non-English language here. What does 4/12 mean under (age of diagnosis) ?
- Table 2 formatting is better but headings should also be adjusted for ease of reading
- Figure 3 has abbreviations which are not clear what they mean. – these should be outlined in a legend.
Round 2
Reviewer 2 Report
Thanks for addressing my suggested changes. The manuscript has improved markedly and is easier to read.
Author Response
Dear Reviewer (2),
Thank you once more for your valuable opinion.
I would like to add that the minor corrections of English language were made by a native English speaker.
With kind regards,
Anna Kutkowska-Kaźmierczak